# Towards Parametric Robust Activation Functions in Adversarial Machine Learning

**Sheila Alemany**[1], **Alberto Dominguez**[2], **Ilan Grapel**[2], **Niki Pissinou**[3]

[1,3]School of Computing and Information Sciences, Florida International University
[2]National Science Foundation Research Experience for Teachers Fellow
[1]salem010@fiu.edu, [3]pissinou@fiu.edu

## Abstract

Machine learning's vulnerability to adversarial perturbations has been argued to stem from a learning model's non-local generalization over complex input data. Given the incomplete information in a complex dataset, a learning model captures non-linear patterns between data points with volatility in the loss surface and exploitable areas of low-confidence knowledge. It is the responsibility of activation functions to capture the non-linearity in data and, thus, has inspired disjointed research efforts to create robust activation functions. This work unifies the properties of activation functions that contribute to robust generalization with the generalized gamma distribution function. We show that combining the disjointed characteristics presented in the literature provides more effective robustness than the individual characteristics alone[1].

## 1 Loss Surface versus Adversarial Example Optimization

Attackers generate adversarial inputs by optimizing their attacks to find a minimal perturbation to an existing input based on the model's probability density function such that it causes incorrect model output Chaubey et al. (2020). As these malicious examples have become increasingly stealthy, literature has heavily focused on increasing the performance accuracy of learning models despite the existence of such imperceptible perturbations (i.e., increasing adversarial robustness). Existing defenses consist of adversarial training Ilyas et al. (2019); Wong et al. (2019); Sriramanan et al. (2021); Kim et al. (2023), regularization Qin et al. (2019), and varying data augmentations/representations Amsaleg et al. (2020); Gong et al. (2021); Alemany & Pissinou (2022) with adversarial training remaining the most effective Lin et al. (2023). A common thread among these proposed improvements is the importance of the input and parameter loss surface that facilitates or inhibits an adversary's ability to effectively attack. Activation functions play a significant role in the created input and parameter loss surface of a learning model, even in scenarios where they have comparable high generalization performance. A simple example is seen in Figure 1.

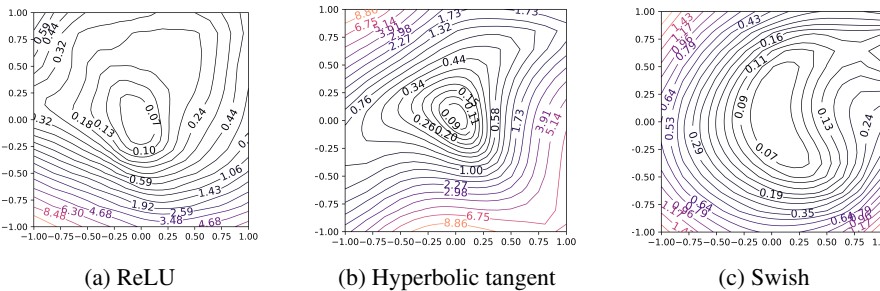

|            |                       |           |
| :--------: | :-------------------: | :-------: |
| (a) ReLU   | (b) Hyperbolic tangent | (c) Swish |

Figure 1: The parameter space loss surface for a simple 2-layer neural network trained to learn $f(x) = x^2$ with 30 data points. Each model was identically trained with changes only present for the activation functions in each layer.

---

[1]The source code of this work can be found at github.com/sheilaalemany/tiny-paper-activation-function

## 2 ACTIVATION FUNCTIONS IN ADVERSARIAL MACHINE LEARNING

Tavakoli et al. (2021) proposed SPLASH, a piecewise dynamic linear function that is optimized for robust generalization during training. Their optimized activation function is **non-monotonic** and aligns with the results by Zhao & Griffin (2016) which proved the importance of **symmetric activations to suppress signals of exceptional magnitude** (i.e., larger perturbations). Rozsa & Boult (2019) introduced tent activation functions with **bounded open space risk** since they observed that adversaries exploit the unbounded open space risk that standard monotonic activation functions provide. Parisi et al. (2020) reached a similar activation shape as Tavakoli et al. (2021) without iterative learning that also improved general robustness. Additionally, Singla et al. (2021) suggested the use of **smoothness and low curvature** in activation functions to increase robust generalization, specifically when using adversarial training to avoid overfitting to adversarial examples. Dai et al. (2022) observed similar effects through their ReBLU function though they did not explain why this was the case.

To address our research goal of improving *parametric robust activation functions*, the following research question needs to be further explored: *what loss surface properties should be encouraged by activation functions to increase adversarial robustness?* We observed that unifying the shape properties of existing activation research efforts through the parametric generalized gamma distribution function provides insight into this research question while improving robustness.

## 3 PARAMETRIC GENERALIZED GAMMA ACTIVATION FUNCTION

The generalized gamma distribution has two shape parameters $(\alpha, c)$, and a scale parameter $(s)$. For the activation function to be continuous and differentiable from $(-\infty, \infty)$, we define the generalized gamma activation function as: $f(x, \alpha, c) = \frac{|c|x^{c\alpha-1}e^{-x^c}}{s\Gamma(\alpha)}$ for $x = \frac{x-\mu}{\beta} \geq 0$, $\alpha > 0$, $c \neq 0$ and $\Gamma(\alpha)$ is the Gamma function on $\alpha$. For $x < 0$, $f(x, \alpha, c) = 0$. We achieved a range within $[0, 1]$ and function shape that met the implications from past robust activation efforts from Section 2 and with the parameters $\alpha = 1$, $c = 3$, $s = 1.17$, $\beta = 3$, and $\mu = -2.6$. Figure 2a shows our generalized gamma activation function compared to other activation functions.

**Evaluation** We tested our activation function on each hidden layer with the MNIST and CIFAR-10 datasets and compared the robustness results to the ReLU, Tanh, Swish and SPLASH activations. Figure 2b and 2c shows that our approach significantly outperforms the other activations for all perturbation budgets, especially for the relatively more complex dataset CIFAR-10. We also observed a 26.3% increase in robustness compared to SPLASH with the CIFAR-10 dataset and a perturbation budget of $e = 0.06$. Similar evaluation results were observed in the Appendix against other attack algorithms.

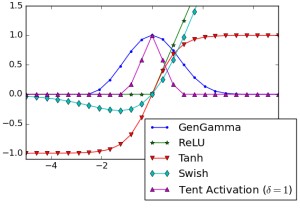
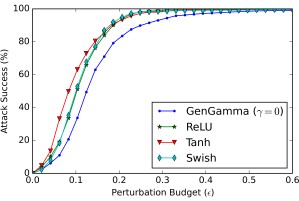
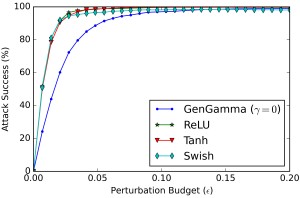

(a) Activation Shape Comparison  (b) FGSM attack on MNIST  (c) FGSM attack on CIFAR-10

Figure 2: Our generalized gamma activation shape comparisons (Figure a) and robustness comparisons with the FGSM Goodfellow et al. (2015) attack on the LeNet-5 architecture (Figure b and c).

**Discussion and Conclusions** The unification of the aforementioned shape properties in our generalized gamma activation function increases robustness because it uniquely encourages a less volatile (or "flatter") parameter loss surface as we stray from the data points Kanai et al. (2023). This suppresses the risk around the low confidence regions that are often exploited by adversaries. Since "flatness" is relative and difficult to achieve with more complex datasets, we observe that parameter loss surface volatility that is concentrated around the most significant features maintains performance accuracy while increasing robustness.

URM STATEMENT

The authors acknowledge that the first author of this work meets the URM criteria of ICLR 2023 Tiny Paper Track.

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

# A APPENDIX

## A.1 INITIALIZATION

For initialization of our activation function, we made sure to avoid the problem of exploding and vanishing gradients during gradient propagation to avoid a false sense of security in our evaluations Athalye et al. (2018). Additionally, similarly to the tent activations, initialization of the generalized gamma activation function needs to ensure that significant inputs do not fall into saturated regions to avoid low model performance Rozsa & Boult (2019).

## A.2 TRAINING

Our training architecture for this work was LeNet-5 with our activation function requiring approximately 10% more epochs to reach comparable performance (within 5%) to the ReLU, hyperbolic tangent, and swish activation functions. The learning rate was 0.01 for all tests. For the attack methods, we used Fast Gradient Sign Method (FGSM) Goodfellow et al. (2015), Projected Gradient Descent (PGD) Madry et al. (2018), and C&W $l_2$ Carlini & Wagner (2017) attack implementations from the Adversarial Robustness Toolbox by IBM Research Nicolae et al. (2019) with no changed hyperparameters.

## A.3 ADDITIONAL EVALUATION RESULTS

|  | **GenGamma** | **ReLU** | **Tanh** | **Swish** |
|---|---|---|---|---|
| C&W $l_2$ attack on MNIST | $88 \pm 2.44$ | $84.5 \pm 0.57$ | $83.5 \pm 1.29$ | $84.87 \pm 0.50$ |
| C&W $l_2$ attack on CIFAR-10 | $30.75 \pm 16.68$ | $22.25 \pm 1.50$ | $10.50 \pm 15.00$ | $21.25 \pm 2.50$ |

Table 1: Carlini & Wagner $l_2$ results on the LeNet-5 architecture in the format: performance percentage $\pm$ standard deviation.

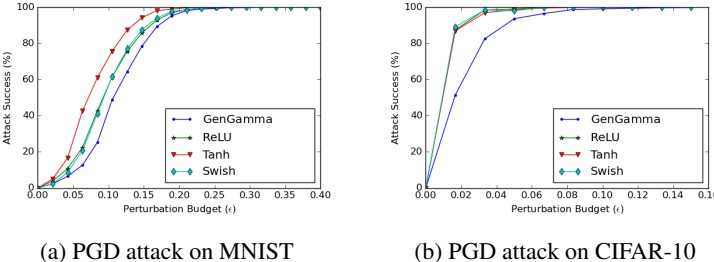

(a) PGD attack on MNIST          (b) PGD attack on CIFAR-10

Figure 3: Projected Gradient Descent attack on the LeNet-5 architecture for varying perturbation budgets ($\epsilon$).

Table 1 and Figure 3 show consistent results for our generalized gamma activation function against the PGD and Carlini & Wagner attacks: improved robustness compared to the ReLU, hyperbolic tangent, and swish activations functions. Unfortunately, we did not compare against tent activations because we could not reach a comparable performance during benign training.

A con that is introduced with the generalized gamma function is an increase in training time. Although the training time was not increased 2-3x times like Tavakoli et al. (2021) and Rozsa & Boult (2019) efforts, our training increased by $25\%$ relative to the ReLU, hyperbolic tangent, and swish activation functions. However, the parametric nature of the generalized gamma activation function allows us to tailor the shape and scale parameters to improve convergence during training in future work.

