# OpenReview forum: "Towards Parametric Robust Activation Functions in Adversarial Machine Learning"
_ICLR.cc/2023/TinyPapers — Submitted to Tiny Papers @ ICLR 2023_

### Official Review · Reviewer_joxW · 2023-03-22

**Confidence:** 5

**Summary Of Contributions:**

This paper leverages the property of activation functions to improve the robustness of adversarial training.

**Rating:**

Needs Clarification (NC): a submission which does not meet the reviewing criteria and needs clarification for its described problem or solution

**Strengths And Weaknesses:**

Strengths:
1. Exploit the shape of the activations for better robustness.

Weaknesses:
1. The idea is not novel and this work cannot contribute to the community. Using different activations for adversarial training have been explored before.
2. The authors have not described why they want to propose the scheme described in the paper.
3. Lack of comparison. Experiments are wrong and very confusing. The authors have not detailed the setup and compared with other works.
4. The written and organization are very bad.


**Suggested Changes:**

See the weaknesses part.

---

### Official Review · Reviewer_jUw5 · 2023-03-28

**Confidence:** 4

**Summary Of Contributions:**

This paper studies the effect of parametric activations in adversarial training. Results show improved robustness against multiple attacks.

**Rating:**

Great Start (GS): a submission which meets some of the reviewing criteria but has room for improvement

**Strengths And Weaknesses:**

Strengths:
- The writing is mostly clear.
- The paper designs a parametric activation function.
- Results show improved robustness against multiple attacks by using the designed activation function.

Weaknesses:
- It is unclear how the models are trained, and if the proposed activation function can improve adversarial training.
- Only some basic gradient-based white-box attacks are considered. I am not sure if there can be an issue of obfuscated gradients [1] and if the robustness still holds against other attacks in [1].
- There are many other works on parametric activations in adversarial training, which are not compared (see https://scholar.google.com/scholar?hl=en&q=PARAMETRIC+ACTIVATION++in+adversarial+training).

[1] Athalye, A., Carlini, N., & Wagner, D. (2018, July). Obfuscated gradients give a false sense of security: Circumventing defenses to adversarial examples. In International conference on machine learning (pp. 274-283). PMLR.


**Suggested Changes:**

Use parenthetical citation (\citep) instead of in many places of the paper, if the author names are not part of the sentence to be read.

---

### Author Response · Authors · 2023-05-30
**Opt-in for archival**

We would like to opt-in for archival for this tiny paper.

All the feedback is highly appreciated! We will be incorporating the feedback provided into the full paper that is pending submission, including the analysis with adversarial training and a more in-depth related work that highlights this effort's novelty.

Thanks again for this great experience!

---

### Comment · Area_Chair_sagL · 2023-06-05
**Ready to archive**

This work meets the threshold for archival, contains the URM statement, and is deanonymized.

---

### Meta-Review · Area_Chair_sagL · 2023-04-07

**Recommendation:** Invite to present
**Confidence:** 4

**Metareview:**

Thank you for your submission. This is an interesting paper with some nice experimental results. However, the reviewers have noted some concern with some necessary related work missing, as well as discussion surrounding how this might impact adversarial training. While additional experiments on black-box attacks would be ideal, these can often take longer and require more compute, and they don't appear to be absolutely necessary for the submission to meet the CCR criteria. Therefore, with the added discussion and related work the revisions should be fairly minor.

**Summary:**

This work introduces a new parametric activation function, called the gamma activation function. In an analysis of this and a variety of other activation functions, they find that their activation function leads to more robust models in the presence of an FGSM adversary..

**Reason For Not Giving A Higher Recommendation:**

- Missing some related work that should be discussed
- Attack appears to perform worse than other activation functions in the presence of stronger attacks (PGD and CWL2 results in the appendix)
- Code not provided

**Reason For Not Giving A Lower Recommendation:**

- Writing is clear
- Experiments have been done under multiple attacks
- Introduces a new parametric activation function

---

### Decision · Program_Chairs · 2023-04-10

Invite to present